# ACCELERATING DNN TRAINING THROUGH SELECTIVE LOCALIZED LEARNING

## ABSTRACT

Training Deep Neural Networks (DNNs) places immense compute requirements on the underlying hardware platforms, expending large amounts of time and energy. We propose LoCal+SGD, a new algorithmic approach to accelerate DNN training by *selectively combining localized or Hebbian learning within a Stochastic Gradient Descent (SGD) based training framework*. Back-propagation is a computationally expensive process that requires 2 Generalized Matrix Multiply (GEMM) operations to compute the error and weight gradients for each layer. We alleviate this by selectively updating some layers' weights using localized learning rules that require only 1 GEMM operation per layer. Further, since the weight update is performed during the forward pass itself, the layer activations for the mini-batch do not need to be stored until the backward pass, resulting in a reduced memory footprint. Localized updates can substantially boost training speed, but need to be used selectively and judiciously in order to preserve accuracy and convergence. We address this challenge through the design of a *Learning Mode Selection Algorithm*, where all layers start with SGD, and as epochs progress, layers gradually transition to localized learning. Specifically, for each epoch, the algorithm identifies a *Localized→SGD* transition layer, which delineates the network into two regions. Layers before the transition layer use localized updates, while the transition layer and later layers use gradient-based updates. The trend in the weight updates made to the transition layer across epochs is used to determine how the boundary between SGD and localized updates is shifted in future epochs. We also propose a low-cost weak supervision mechanism by controlling the learning rate of localized updates based on the overall training loss. We applied LoCal+SGD to 8 image recognition CNNs (including ResNet50 and MobileNetV2) across 3 datasets (Cifar10, Cifar100 and ImageNet). Our measurements on a Nvidia GTX 1080Ti GPU demonstrate upto $1.5\times$ improvement in end-to-end training time with $\sim0.5\%$ loss in Top-1 classification accuracy.

## 1 INTRODUCTION

Deep Neural Networks (DNNs) have achieved continued success in many application domains involving images (Krizhevsky et al., 2017), videos (Ng et al., 2015), text (Zhou et al., 2015) and natural language (Goldberg & Hirst, 2017). However training state-of-the-art DNN models is computationally quite challenging, often requiring exa-FLOPs of compute as the models are quite complex and need to be trained using large datasets. Despite rapid improvements in the capabilities of GPUs and the advent of specialized accelerators, training large models using current platforms is still quite expensive and often takes days to even weeks. In this work, we aim to reduce the computational complexity of DNN training through a new algorithmic approach called LoCal+SGD[1], which alleviates the key performance bottlenecks in Stochastic Gradient Descent (SGD) through *selective use of localized or Hebbian learning*.

**Computational Bottlenecks in DNN Training.** DNNs are trained in a supervised manner using gradient-descent based cost minimization techniques such as SGD (Bottou, 2010) or Adam (Kingma & Ba, 2015). The training inputs (typically grouped into minibatches) are iteratively forward propagated ($FP$) and back propagated ($BP$) through the DNN layers to compute weight updates that push the network parameters in the direction that decreases the overall classification loss.

---

[1]In addition to combining localized and SGD based learning, LoCal+SGD is Low-Calorie SGD or SGD with reduced computational requirements

Back-propagation is computationally expensive, accounting for 65-75% of the total training time on GPUs. This is attributed to two key factors: (i) $BP$ involves 2 Generalized Matrix Multiply (GEMM) operations, one to propagate the error across layers and the other to compute the weight gradients, and (ii) when training on distributed systems using data/model parallelism(Dean et al., 2012b; Krizhevsky et al., 2012), aggregation of weight gradients/errors across devices incurs significant communication overhead. Further, $BP$ through auxiliary ops such as batch normalization are also more expensive than $FP$.

**Prior Efforts on Efficient DNN Training.** Prior research efforts to improve DNN training time can be grouped into a few directions. One group of efforts enable larger scales of parallelism in DNN training through learning rate tuning (You et al., 2017a; Goyal et al., 2017; You et al., 2017b) and asynchronous weight updates (Dean et al., 2012a). Another class of efforts employ importance-based sample selection during training, wherein 'easier' training samples are selectively discarded to improve runtime (Jiang et al., 2019; Zhang et al., 2019). Finally, model quantization (Sun et al., 2019) and pruning (Lym et al., 2019) can lead to significant runtime benefits during training by enabling the use of reduced-bitwidth processing elements.

LoCal+SGD**: Combining SGD with Localized Learning.** Complementary to the aforementioned efforts, we propose a new approach, LoCal+SGD, to alleviate the performance bottlenecks in DNN training, while preserving model accuracy. Our hybrid approach combines Hebbian or localized learning (Hebb) with SGD by selectively applying it in specific layers and epochs. Localized learning rules (Hebb; Oja, 1982; Zhong, 2005) utilize a single feed-forward weight update to learn the feature representations, eschewing $BP$. Careful formulation of the localized learning rule can result in $\sim 2\times$ computation savings compared to SGD and also significantly reduces memory footprint as activations from $FP$ needed not be retained until $BP$. The reduction in memory footprint can in turn allow increasing the batch size during training, which leads to further runtime savings due to better compute utilization and reduced communication costs. It is worth noting that localized learning has been actively explored in the context of unsupervised learning (Chen et al., 2020; van den Oord et al., 2018; Hénaff et al., 2019). Further, there has been active research efforts on neuro-scientific learning rules (Lee et al., 2015; Nøkland, 2016). Our work is orthogonal to such efforts and represents a new application of localized learning in a fully supervised context, wherein we selectively combine it within an SGD framework to achieve computational savings.

Preserving model accuracy and convergence with LoCal+SGD requires localized updates to be applied judiciously *i.e.,* only to selected layers in certain epochs. We address this challenge through the design of a *learning mode selection algorithm*. At the start training, the selection algorithm initializes the learning mode of all layers to SGD, and as training progresses determines the layers that transition to localized learning. Specifically, for each epoch, the algorithm identifies a *Localized→SGD* transition layer, which delineates the network into two regions. Layers before the transition layer use localized updates, while subsequent layers use gradient-based updates. This allows $BP$ to stop at the transition layer, as layers before it have no use for the back-propagated errors. The algorithm takes advantage of the magnitude of the weight updates of the *Localized→SGD* transition layer in deciding the new position of the boundary every epoch. Further, we provide weak supervision by tweaking the learning rate of locally updated layers based on overall training loss.

**Contributions**: To the best of our knowledge, LoCal+SGD is the first effort that combines localized learning (an unsupervised learning technique) within a supervised SGD context to reduced computational costs while maintaining classification accuracy. This favorable tradeoff is achieved by LoCal+SGD through a Learning Mode Selection Algorithm that applies localized learning to selected layers and epochs. Further improvement is achieved through the use of weak supervision by modulating the learning rate of locally updated layers based on the overall training loss. Across 8 image recognition CNNs (including ResNet50 and MobileNet) and 3 datasets (Cifar10, Cifar100 and ImageNet), we demonstrate that LoCal+SGD achieves up to $1.5\times$ improvement in training time with $\sim 0.5\%$ Top-1 accuracy loss on a Nvidia GTX 1080Ti GPU.

## 2 LoCal+SGD: Combining SGD with Selective Localized Learning

The key idea in LoCal+SGD is to apply localized learning to selected layers and epochs during DNN training to improve the overall execution time, without incurring loss in accuracy. The following components are critical to the effectiveness of LoCal+SGD:

$W_L$: Filter Weight of Layer L
$z_L$: Pre-Activation o/p of Layer L
$a_L$: Post-Activation o/p of Layer L
$\delta_L$: Error at Layer L
$\eta$: Learning Rate

| Stage | Localized Updates | | SGD-Based Updates | |
|---|---|---|---|---|
| FP | $z_l = GEMM(a_{l-1}, W_l)$ | (2) | | |
| Update Stage | $\Delta W_l = GEMM(a_{l-1}, z_l)$ $\qquad$ (3) $\qquad$ $W_l = W_l + \eta \dfrac{\Delta W_l}{\|\Delta W_l\|}$ $\qquad$ (4) | | $\delta_l = GEMM(\delta_{l+1}, W_l)$ $\qquad$ (5) $\quad\ $ $\Delta W_l = GEMM(a_{l-1}, \delta_l)$ $\qquad$ (6) $\qquad\ $ $W_l = W_l + \eta \Delta W_l$ $\qquad\quad\ $ (7) | |

NOTE: GEMM includes operations in convolutional and fully-connected layers

Figure 1: Comparing Localized Updates and SGD-based $BP$

- **Localized Learning Rule Formulation.** We formulate a computationally efficient localized learning rule and highlight the clear runtime benefits when compared to SGD.

- **Learning Mode Selection Algorithm.** We propose a learning mode selection algorithm that chooses between localized learning and SGD-based learning for each layer in every epoch, based on the potential impact on accuracy and computational benefits.

- **Weak Supervision.** We propose a weak supervision technique, which comprises of a low-cost supervision signal communicated to the localized learning layers in each epoch. The signal modulates the learning rates of these layers based on the rate of change of the overall classification loss.

In the following sub-sections, we describe the salient aspects of these components in greater detail.

## 2.1 EFFICIENT LOCALIZED LEARNING

Localized learning has been extensively explored in the context of unsupervised learning, demonstrating success on small (<= 3 layer) networks using relatively simpler datasets (*e.g.* MNIST, Cifar-10) (LeCun & Cortes, 2010; Krizhevsky et al., a)) with an accuracy gap that is yet to be bridged on larger datasets (*e.g.* ResNet50 or MobileNetV2 on ImageNet (Deng et al., 2009)). First proposed in (Hebb), the key intuition behind localized learning rules is to encourage correlations between neurons that have similar activation patterns. Equation 1 depicts the Hebbian weight update proposed in (Hebb), for a synapse with weight $W$, connecting a pair of input and output neurons whose activation values are represented by $x$ and $y$ respectively, with $\eta$ as the learning rate.

$$\triangle W = \eta \cdot x \cdot y \qquad (1)$$

Considerable research has gone into evolving this equation over the years to improve the performance of localized learning (Oja, 1982; Zhong, 2005). However, many of the proposed rules are computationally complex, or are difficult to parallelize on modern hardware platforms such as GPUs and TPUs. Since our primarily goal is improving DNN training time, we adopt the computationally simple localized learning rule presented in Equation 1.

The learning rule in Equation 1 assumes a distinct synapse between each input and output neuron pair. While its application to fully-connected (fc) layers is straightforward, we need to consider the sharing of weights between neuron pairs in convolutional (conv) layers. For updating a shared weight of a conv layer, we calculate the individual updates due to each pair of pre- and post-synaptic neurons sharing the weight and sum all such updates. This essentially reduces to a convolution operation between the input and output activations of the layer and can be expressed by Equation 3 in Figure 1. For further computational efficiency improvement, unlike Equation 1, we consider the pre-activation-function values of the outputs i.e., $z_l$ instead of their post activation value $a_l$. Further, we normalize the localized update values as shown in Equation 4 of Figure 1, as it was observed to achieve better convergence in practice.

Overall, we utilize Equations 3 and 4 from Figure 1 to perform the weight updates in all layers that are earlier than the *Localized→SGD* transition layer during a certain epoch. All other layers continue to be updated using SGD-based $BP$, expressed by Equations 5-7 in Figure 1. SGD updates are applied to batch-normalization layers present after the *Localized→SGD* transition layer, and are otherwise skipped. Clearly, Equation 3 has the same computational complexity as Equation 6 of SGD-based $BP$ for conv and fc layers. Thus, from Figure 1, we can directly infer that our localized learning rule will be considerable faster than SGD-based $BP$. In practice, we measured this improvement to be more than $2\times$ on a NVIDIA GTX 1080Ti GPU for the ImageNet-ResNet50 benchmark, across all conv and fc layers. In addition to the computational complexity, the memory footprint of SGD-based

$BP$ is also higher. This is because DNN software frameworks commonly store all activation values computed during $FP$ to avoid recomputing $a_{l-1}$, the input activations to the layers, used in Equation 6 of SGD-based $BP$. In contrast, the localized update for a layer can be performed as soon as the $FP$ through the layer is complete. The activation tensor $a_l$ of layer $L$ can be discarded or over-written as soon as $FP$ proceeds to the next layer in the network, thereby freeing up a significant portion of on-device memory during training. In turn, this can allow larger minibatch sizes to be accommodated on a given hardware platform, when the localized updates are applied on a sufficient number of layers.

## 2.2 LEARNING MODE SELECTION ALGORITHM

The compute benefits of localized learning come at the cost of potential loss in classification accuracy with respect to SGD training. Thus, we utilize a learning mode selection algorithm to judiciously choose when and where to apply localized learning. The proposed algorithm identifies the learning mode of each layer at every epoch to maximize the runtime benefits, while incurring minimal losses in classification accuracy.

To design an efficient learning mode selection algorithm, we first study the effects of different spatio-temporal patterns of localized learning on the computational efficiency and classification accuracy of a network. We specifically investigate whether localized learning is more suitable for specific layers in the network and specific phases in the training process.

*Impact on runtime efficiency*: We first analyze the spatial trends, *i.e.*, if locally updating specific layers in the network results in better runtime efficiency. In a particular epoch, if a convolutional layer $L$, updated with SGD precedes a convolutional layer $K$, that is updated locally, calculating the SGD-based error gradients of Layer $L$, i.e. $\delta_L$, requires error propagation through the locally updated layer $K$. From a compute efficiency perspective, the benefits of using localized-updates in layer $K$ completely vanish. Thus, it makes sense to partition the network into two regions - a prefix (set of initial layers) that are updated using localized learning, followed by layers that are updated with SGD. SGD-based $BP$ is stopped at the junction of the two regions. Naturally, the compute benefits increase when the number of locally updated layers are higher and thus the boundary i.e., the *Localized→SGD* transition layer is moved deeper into the network.

The impact of different temporal patterns on runtime efficiency is quite straightforward, with higher number of locally updated epochs leading to higher benefits. Further, as the compute complexity of localized updates is constant across different epochs, these benefits are agnostic of which particular epoch involves localized learning.

*Impact on accuracy*: To analyze the impact on accuracy, we first examine the nature of features learnt by different layers trained by SGD. It is commonly accepted that the initial layers of a network (Agrawal et al., 2014) perform feature extraction, while later layers aid in the classification process. As localized learning demonstrates better performance for feature extraction, applying it more aggressively, *i.e* for higher number of epochs, in the initial layers has a much smaller impact accuracy. However, for later layers in the network, the number of localized learning epochs should be progressively reduced to preserve accuracy.

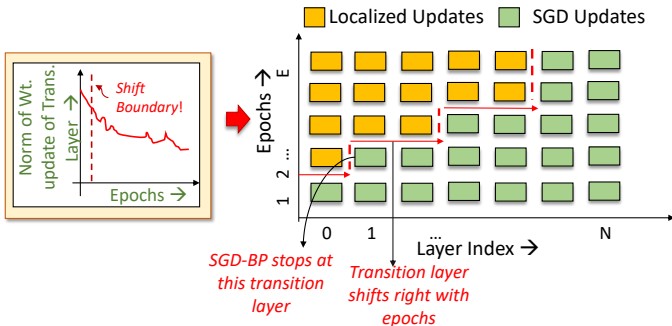

Figure 2: Overview of the Learning Mode Selection Algorithm

Overall, based on the impact of localized learning on both runtime and accuracy, we find that a good learning mode selection algorithm should favor application of localized learning to a contiguous group of initial layers, while ensuring fewer or no localized learning epochs in later layers. We further

impose an additional constraint on top of this spatio-temporal pattern. Specifically, we allow each layer to transition from one learning mode to another at most once during the entire training process. We empirically observe that utilizing SGD as the initial learning mode allows the network to achieve a higher accuracy than utilizing localized learning as the initial mode. SGD essentially provides a better initialization point for all layers, and the subsequent use of localized updates enables the training to converge with good accuracy.

In accordance with the above considerations, we propose a learning mode selection algorithm, described in Algorithm 1, that identifies the position of the boundary or the *Localized→SGD* transition layer every epoch. To that end, the algorithm analyzes the $L_2$ norm of the SGD weight updates made to the *Localized→SGD* transition layer across epochs and determines whether the boundary can be shifted deeper into the network for the next epoch. In order to ensure stability in the training process, the algorithm moves the boundary at most once in every K epochs. It calculates the running average of the norm of the updates, $W_{avg}$, over the last $K$ epochs (line 1). The boundary is shifted to the right only if the weight update in epoch $E$ is within a fraction $\alpha$ of $W_{avg}$, and K epochs have transpired since the last transition (line 2). The rationale for this criterion is that sustained high magnitudes of weight updates in the transition layer indicate that they are potentially critical to accuracy, in which case the transition layer must continue being updated with SGD. If the criterion is not satisfied, the boundary remains stationary (line 5).

---

**Algorithm 1** Learning Mode Selection Algorithm

**Input:** $T_E$ (Index of the transition layer at epoch E), $e_k$ (epochs since last transition), $|| \triangle W_E ||$ ($L_2$ norm of the weight update of the transition layer at epoch E), $K$ (minimum interval between transitions), $t_{shift}$ (number of layers to shift boundary)

**Output:** $T_{E+1}$ (Index of the transition layer at epoch E+1)

1: $W_{Avg} = \frac{1}{K} \sum_{e=E-K}^{e=E-1} || \triangle W_e ||$
2: **if** $|| \triangle W_E || <= \alpha \cdot W_{Avg}$ **and** $e_k$>=K
3:     $T_{E+1} = T_E + t_{shift}$
4:     $e_k = 0$
5: **else**
6:     $T_{E+1} = T_E$
7:     $e_k = e_k + 1$

---

The value of $\alpha$ is set by analyzing the trends in the weight update magnitudes across the training process for different networks. The hyper-parameter $t_{shift}$ is set to the size of a recurring block, such as the residual blocks in ResNets and MobileNetV2. The hyper-parameter K is selected in a manner that ensures that localized updates are never applied beyond some fraction of the initial network layers. We denote this fraction as $L_{max}$, and is set to 0.75 in all our experiments. Equation 2 is used to compute K for a network of L layers and a total training period of $E_{max}$ epochs.

$$K = \frac{E_{max}}{L_{max} * \frac{L}{t_{shift}}} \qquad (2)$$

Figure 3: Progression of *Localized→SGD* transition layer

In Figure 3, we plot the progression of the transition layer across the ResNet-34 and -50 benchmarks trained on the ImageNet dataset using LoCal+SGD. Interestingly, the weight update norm metric automatically modulates the rate at which the boundary progresses, as the boundary traverses the deeper layers at a slower rate.

## 2.3 WEAK SUPERVISION

To further bridge the accuracy gap between our approach and end-to-end SGD training, we introduce weak supervision in the locally updated layers. Unlike the SGD-updated layers, the locally updated layers in our approach cannot take advantage of the information provided by supervision, i.e., the classification error evaluated at the output. We utilize this supervised information through a low-cost weak supervision scheme that consists of a single signal sent to all layers updated locally in a particular epoch, and is derived from the classification loss observed over past few epochs. The weak supervision scheme is described in Algorithm 2.

The key principle behind the weak supervision scheme is to control the learning rates of the locally updated layers based on the rate at which the overall classification loss changes. For example, if the overall classification loss has increased across consecutive epochs, we reverse the direction of the

updates (line 3) in the next epoch. In contrast, the update direction is maintained if the overall loss is decreasing (line 5). We find that this weak supervision provides better accuracy results than other learning rate modulation techniques for the locally updated layers such as Adam or momentum-based updates.

We would like to highlight that traditional SGD provides fine-grained supervision and involves evaluating the error gradients for every neuron in the network. In contrast, the proposed weak supervision scheme provides coarse-grained supervision by forcing all weights to re-use the same loss information. Overall, our weak supervision scheme is not developed with the intent to compete with SGD updates, but is rather a simple, approximate and low-cost technique that brings the final accuracy of LoCal+SGD at par with end-to-end SGD training performance.

---

**Algorithm 2** Weak Supervision Scheme

**Input:** $L_i$ (Overall classification loss at epoch $i$), $lr_L$ (original learning rate of layer $L$)
**Output:** $W_L$ (Weight update of layer $L$)
1: $\triangle W_L = conv(a_{l-1}, z_l)$
2: **if** $L_{i-1} < L_i$
3: $\quad W_L = W_L - lr_L \cdot \frac{\triangle W_L}{||\triangle W_L||}$
4: **else**
5: $\quad W_L = W_L + lr_L \cdot \frac{\triangle W_L}{||\triangle W_L||}$

---

## 3 EXPERIMENTAL RESULTS

In this section, we present the results of our experiments highlighting the compute benefits achieved by LoCal+SGD. We evaluate the benefits across a suite of 8 image-recognition DNNs across 3 datasets. We consider the ResNet18 (He et al., 2015) and VGG13 (Simonyan & Zisserman, 2015) networks for the Cifar10 (Krizhevsky et al., a) and Cifar100 (Krizhevsky et al., b) datasets; and the ResNet34, ResNet50 (He et al., 2015) and MobileNetV2 (Sandler et al., 2018) networks for the ImageNet dataset (Deng et al., 2009). All experiments are conducted on Nvidia GTX 1080Ti GPUs with the batch size set to 64 per GPU, unless otherwise mentioned. Further experimental methodology details for the baseline and proposed approach are provided in the Appendix.

### 3.1 SINGLE GPU EXECUTION TIME BENEFITS

**ImageNet**: Table 1 presents the performance of the baseline (end-to-end SGD training) and the proposed LoCal+SGD algorithm on the ImageNet benchmarks in terms of the Top-1 classification error and runtime observed on a single GPU. For all benchmarks listed here, LoCal+SGD applies localized updates for nearly 50-60% of the layers. As can be seen, LoCal+SGD achieves upto $\sim1.4\times$ reduction in runtime compared to to the baseline, while sacrificing $<0.5\%$ loss in Top-1 accuracy.

Table 1 also compares the performance of LoCal+SGD against existing research efforts designed to improve training efficiency. We perform this analysis against two efforts, namely (i) Training with stochastic depth (Huang et al., 2016) and (ii) Structured Pruning during Training (Lym et al., 2019). Training with stochastic depth, as the name suggests, stochastically bypasses residual blocks by propagating input activations/error gradients via identity or downsampling transformations, resulting in improved training time. However, the approach is targeted towards extremely deep networks and

Table 1: ImageNet

| Network | Training Strategy | Top-1 Error | Speed-Up |
|---|---|---|---|
| ResNet34 | Baseline SGD | 26.6% | $1\times$ |
| | LoCal+SGD | **27.04%** | **1.26$\times$** |
| | Training with Stochastic Depth | 27.89% | 1.13$\times$ |
| | Freezing layers during training | 27.42% | 1.38$\times$ |
| ResNet50 | Baseline SGD | 24.02% | $1\times$ |
| | LoCal+SGD | **24.41%** | **1.36$\times$** |
| | Training with Stochastic Depth | 26.76% | 1.08$\times$ |
| | Pruning during training | 24.89% | 1.32$\times$ |
| | Freezing layers during training | 25.64% | 1.44$\times$ |
| MobileNetV2 | Baseline SGD | 28.41% | $1\times$ |
| | LoCal+SGD | **28.94%** | **1.31$\times$** |
| | Training with Stochastic Depth | 30.53% | 1.17$\times$ |
| | Freezing layers during training | 29.49% | 1.52$\times$ |

as seen in Table 1, it incurs a noticeable accuracy loss on networks such as ResNet34, ResNet50 and MobileNetV2. Compared to training with stochastic depth, our proposal clearly achieves better accuracy as well as training runtime benefits. The key principle behind the pruning during training approach is to reduce the size of the weight and activation tensors in a structured manner during training, thereby providing speed-ups on GPU/TPU platforms. However, on complex benchmarks such as ResNet50, such techniques achieve speed-ups at the cost of significant drop in accuracy ($\sim 1.5\%$). To further demonstrate the utility of localized updates in our approach, we consider a third technique, wherein layers selected to be updated locally for a given epoch are instead frozen, i.e., the parameters are held fixed during that epoch. While this achieves better runtime savings, it incurs considerably higher loss ($\sim 1\%$) in accuracy, further underscoring the benefits of LoCal+SGD.

**CIFAR-10 and CIFAR-100**: Table 2 presents the accuracy and corresponding compute benefits of the baseline and the proposed technique, as well as training with stochastic depth and layer freezing, for the CIFAR-10 and CIFAR-100 datasets. Stochastic depth is applicable only to residual blocks and is hence not considered for the VGG-13 network. Across benchmarks, we observe upto a $1.51\times$ improvement in training runtime. Compared to the ImageNet benchmarks, LoCal+SGD applies localized updates more aggressively in the CIFAR-10 and CIFAR-100 benchmarks i.e., for more layers are updated locally for a higher number of epochs. This leads to the superior compute benefits of the proposed scheme on these benchmarks.

Table 2: Cifar10 and Cifar100

| Network(Dataset) | Training Strategy | Top-1 err. | Speed-Up |
|---|---|---|---|
| ResNet18 (Cifar10) | Baseline SGD | 6.06% | 1× |
| | **LoCal+SGD** | **6.23%** | **1.51×** |
| | Training with Stochastic Depth | 6.79% | 1.35× |
| | Freezing layers during training | 6.51% | 1.65× |
| VGG13 (Cifar10) | Baseline SGD | 7.16% | 1× |
| | **LoCal+SGD** | **7.25%** | **1.31×** |
| | Freezing layers during training | 7.43% | 1.42× |
| ResNet18 (Cifar100) | Baseline SGD | 23.39% | 1× |
| | **LoCal+SGD** | **23.63%** | **1.44×** |
| | Training with Stochastic Depth | 23.97% | 1.35× |
| | Freezing layers during training | 23.74% | 1.62× |
| VGG13 (Cifar100) | Baseline SGD | 31.36% | 1× |
| | **LoCal+SGD** | **31.59%** | **1.32×** |
| | Freezing layers during training | 31.94% | 1.42× |

## 3.2 EXECUTION TIME BENEFITS FOR MULTI-GPU TRAINING

We analyze the memory footprint of the ResNet50 network when trained with LoCal+SGD on the ImageNet dataset. Training first commences with all layers updated with SGD, resulting in a high memory footprint. Due to the 10 GB capacity of the chosen GPU, the mini-batch size is set to 64 per GPU. As the *Localized→SGD* transition layer progresses across the network, the memory footprint required also gradually reduces across epochs. We take advantage of this reduction in memory footprint in the context of distributed training using 4 GPUs with data parallelism. Specifically, we extract additional runtime benefits by increasing the batch size on each GPU,

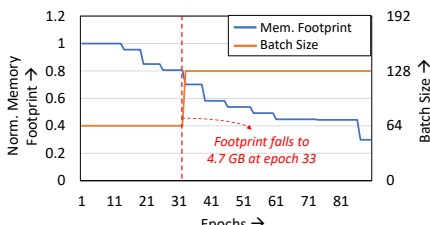

Figure 4: Analyzing memory footprint and batch-size variation

which reduces the frequency of gradient aggregation between devices and alleviates the communication overhead. At epoch 33, the memory footprint per GPU reduces to less than 5 GB, allowing training with an increased mini-batch size of 128 per GPU from epoch 33 onwards. The doubling of the batch-size provides an additional 6% runtime improvement, when measured across the entire training period. We note that other training techniques such as training with stochastic depth cannot exploit this feature, due to minimal reduction in memory footprint.

## 3.3 ABLATION ANALYSIS

As mentioned in Section 2, the hyper-parameters $\alpha$, $t_{shift}$ and $L_{max}$ control the progression of the boundary across the network. Different values of either parameter can result in dif-

Table 3: Analyzing Impact of Increasing Batch-Size on ImageNet

| Network | Training Strategy | Top-1 err. | Speed-Up |
|---------|-------------------|------------|----------|
| | Baseline SGD (fixed batch-size) | 24.06% | 1× |
| ResNet50 | LoCal+SGD (fixed batch-size) | **24.48%** | **1.27×** |
| | LoCal+SGD (variable batch-size) | **24.51%** | **1.34×** |

ferent learning mode configurations during training, resulting in different points in the computational efficiency *vs.* accuracy trade-off space. To understand the trade-off space between accuracy and runtime benefits, we now individually study the impact of each parameter.

*Impact of $\alpha$* : Figure 5 depicts the best compute benefits achieved for different $\alpha$, for accuracy losses ranging from 0.1%-1.5% for the ResNet50 and MobileNetV2 benchmarks on ImageNet. On the ResNet50 benchmark, even while limiting the loss in accuracy 0.1%, LoCal+SGD achieves 1.1× speedup over traditional SGD. The speedups increase to 1.38×-1.47× when around 1.5% loss in accuracy is tolerable.

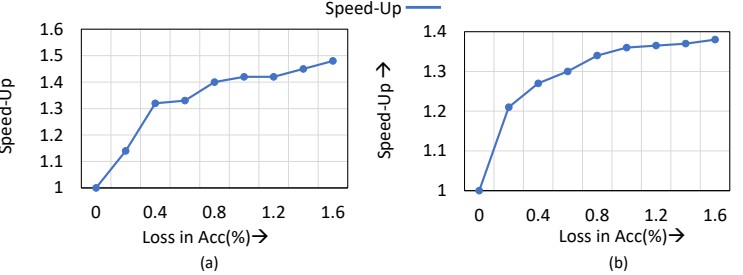

Figure 5: Compute efficiency *vs.* accuracy trade-off on the ImageNet dataset for a) ResNet50 and b) MobileNetV2

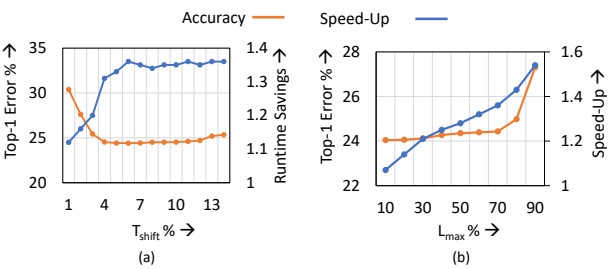

Figure 6: Impact of (a) $t_{shift}$ and (b) $L_{max}$ on accuracy and runtime savings on the ImageNet dataset for ResNet50

*Impact of $t_{shift}$* : Figure 6(a) depicts the accuracy achieved for different $t_{shift}$ values for the ResNet50 network trained using ImageNet. Here, we denote $t_{shift}$ as a percentage of the total network depth. For each value of $t_{shift}$, we identify $\alpha$ and $L_{max}$ that result in the best accuracy while providing considerable runtime savings (at least greater than 10%). The graph indicates that accuracy is severely impacted for extremely small values of $t_{shift}$ that are less than 3%. The accuracy is largely stable in the regime of $t_{shift}$ between 5-12%, and begins to experience small degradations again when $t_{shift}$ exceeds 12%. These trends can be explained by analyzing the rate at which the transition layer progresses, and the number of layers transitioning to localized updates in an epoch for different $t_{shift}$ values. Smaller values of $t_{shift}$ (<3%) give rise to low values of $k$ (∼1-2 epochs), the minimum number of epochs that must elapse before the transition layer can shift again. This results in fast progression of the transition layer across the network, leading to rapid changes in the learning mode at the boundary, thereby negatively impacting accuracy. In contrast, while larger $t_{shift}$ values (>12%) encourage slow progression of the boundary, a larger number of layers transition from SGD to localized updates in a single epoch, thereby impacting performance. We note here that in both cases, while $\alpha$ and $L_{max}$ can be tuned to control the progression and mitigate the loss in accuracy, the runtime savings is vastly reduced (<10%). Furthermore, for fixed values of $L_{max}$ and $\alpha$, $t_{shift}$ is largely insensitive to runtime benefits, as the average number of layers updated with localized updates remains similar. Hence, for best accuracy and runtime benefits we set $t_{shift}$ in the range of 5-10% for all networks.

*Impact of $L_{max}$* : Figure 6(b) depicts the impact of $L_{max}$ on accuracy for the ResNet50 network. For each $L_{max}$, we identify the $\alpha$ and $t_{shift}$ that provide the best runtime benefits with minimal loss in accuracy (less than 0.5%). As with $t_{shift}$, we denote $L_{max}$ as a percentage of the total network

depth. As seen in the figure, the degradation in accuracy increases slowly for $L_{max}$ in the initial layers - it is merely 0.1% at around $L_{max} = 30\%$, and increases to 0.4-0.5% for $L_{max} = 60$-70%. However, the accuracy degradation sharply increases beyond 2% once $L_{max}$ exceeds 90% of the network depth. Further, runtime benefits generally increase with higher values of $L_{max}$, for fixed $t_{shift}$ and $\alpha$. Hence, for achieving a good accuracy versus runtime trade-off, we usually set $L_{max}$ to 75% for all networks.

## 4 RELATED WORK

This section discusses related research efforts to the proposed LoCal + SGD training technique. These efforts can be broadly categorized into two classes. The first class of efforts focus on compute efficient DNN training. All efforts belonging to this class utilize gradient-descent algorithms to train the DNN model. These techniques are largely complementary to LoCal+SGD, as they can potentially be applied to the parts of the DNN model updated with SGD. In Section 3, we demonstrated how LoCal+SGD achieves superior accuracy versus computational efficiency trade-off than some of these efforts. Further, the second class of efforts involve neuro-scientific faithful learning rules, such as feedback alignment based efforts etc (Nøkland, 2016). Our work is orthogonal to such efforts, as we selectively combine localized learning rules with SGD for better computational efficiency.

We elucidate upon the different research efforts in both directions as follows.

**Hyper-parameter tuning**: Many notable algorithmic efforts are directed towards achieving training efficiency by controlling the hyper-parameters involved in gradient-descent, notably the learning rate. (You et al., 2017a; Akiba et al., 2017; Goyal et al., 2017; You et al., 2017b) propose learning rate tuning algorithms that achieve training in less than an hour with no loss in accuracy, when distributed to over hundreds of CPU/GPU cores.

**Model size reduction during training**: Model size reduction via pruning and quantization is a popular technique to reduce compute costs during inference. In many of these efforts, a dense or full precision model is re-trained or fine-tuned to obtain a pruned or quantized model. Recently, several efforts have also investigated dynamically pruning (Lym et al., 2019) or quantizing (Sun et al., 2019) a model during training itself. The reduction in model size results in training speed-ups. Taking a slightly different approach (Huang et al., 2016) proposes stochastically dropping residual blocks on extremely deep networks such as ResNet-1202, not only for training runtime benefits but also better accuracies due to improved gradient strength.

**Instance importance based training**: Recent research efforts have discovered that not all training samples are required for improving loss minimization during SGD training (Jiang et al., 2019; Zhang et al., 2019). That is, a sizable fraction of the samples can be skipped during several epochs, depending on their impact on the classification loss evaluated during $FP$. This translates to a reduction in mini-batches, providing considerable runtime benefits.

**Neuro-scientific learning rules**: Back-propagation algorithms utilized in DNN training are not biologically plausible, and do not explain how learning actually happens in the brain. To this end, there have been several efforts that develop biological faithful learning algorithms, and demonstrate considerable success on complex benchmarks including Cifar10 and ImageNet. For example, unlike conventional DNN training, feedback alignmnent algorithms (Nøkland, 2016) tackle the weight transport problem (Liao et al., 2015) by allowing for asymmetry in the weight values during forward and back propagation. Likewise, Target-Propagation (Lee et al., 2015) encourages neural activity to reach desired target activations evaluated during forward propagation itself, instead of utilizing loss gradients.

## 5 CONCLUSION

In this paper, we introduce a new approach to improve the training efficiency of state-of-the-art DNNs. Specifically, we take advantage of the computationally efficient nature of localized learning rules and selectively update some layers with these rules instead of SGD. We design an intelligent learning mode selection algorithm that determines the update method for the convolutional layers of the network in every epoch while maintaining the accuracy level and extracting maximum benefits. Further, we also implement a low-cost weak supervision scheme that brings the accuracy of the proposed scheme closer to traditional SGD training. Across a benchmark suite of 8 DNNs, we achieve upto $1.5\times$ reduction in training times, as measured on a modern GPU platform.

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

# 6 APPENDIX

## 6.1 EXPERIMENTAL SETUP

This subsection describes the experimental setup used for realizing the baseline and proposed LoCal+SGD training schemes, on the benchmarks specified in Section 3 of the main paper. We conduct our experiments on the complete training and test datasets of each benchmark, using the PyTorch (Paszke et al., 2019) framework.

**Baseline**: We consider end-to-end SGD training as the baseline in our experiments. The hyper-parameters used in SGD training of each of the benchmarks are described below.

ImageNet: For experiments in Section 3.1 we utilize a batch-size of 64 per GPU, for all benchmarks. For the ResNet50 and ResNet34 benchmarks the initial learning rate set to 0.025. The learning rate is decreased by 0.1 every 30 epochs, for a total training duration of 90 epochs, and the weight decay is $4e - 5$. The MobileNetV2 benchmark utilizes an initial learning rate of 0.0125. We use a cosine learning rate decay schedule, as in (Li et al., 2019) for 150 epochs. The weight decay is set to $4e - 5$. Both benchmarks use an input size of 224*224*3.

For the experiments in Section 3.2, the total batch-size at epoch 1 is 256 (64*4), with the initial learning rate set to 0.1 for the ResNet benchmarks and 0.05 for the MobileNetV2 benchmark. All other parameters remain the same.

Cifar10 and Cifar100: All Cifar10 and Cifar100 experiments utilize a batch-size of 64. The Cifar10 benchmarks are trained with an initial learning rate of $0.05$ that is decayed by 0.1 every 10 epochs, across 90 epochs. The initial learning rate of the Cifar100 benchmarks is $0.025$ and decayed by 0.5 every 20 epochs, for 150 epochs in total. The weight decay is set to $5e - 4$. Both benchmarks utilize an input size of 32*32*3.

LoCal+SGD: In the proposed LoCal+SGD training scheme, the layers updated with SGD are trained with the same hyper-parameters used in the baseline implementation. Further, LoCal+SGD training is conducted using the same number of epochs as baseline SGD training. When a layer is updated locally, the initial learning rate is $0.01$ and is decayed by a factor of 2 and 10 every 30 epochs, for the Cifar and the ImageNet benchmarks respectively. In all experiments, the $\alpha$ parameter is set to 0.95. We measure the accuracy and runtime of the proposed scheme for the same number of training epochs as the baseline implementations.

## 6.2 HYPER-PARAMETER TUNING

To realize LoCal+SGD, we introduce three hyper-parameters: $\alpha$, $t_{shift}$ and $L_{max}$. $t_{shift}$ controls the number of layers that switch to SGD-based updates every epoch, $L_{max}$ is the maximum number of layers that can be updated with localized learning rules, and $\alpha$ determines the position of the

transition layer every epoch by analyzing the gradient information at the boundary between the localized and SGD updates.

To obtain optimized values for these hyper-parameters, we first perform simple grid search using a single network for a particular dataset (for example, we choose the ResNet50 network for ImageNet). We transfer the same hyper-parameter values to other networks for the same dataset. We justify our use of common hyper-parameter values by the following experiment. In Table 4 below, we depict the results on other ImageNet benchmarks (ResNet34 and MobileNetV2) when hyper-parameter tuning is performed for each benchmark individually. As can be seen, the accuracy and runtime benefits are only marginally better than those obtained using a common set of hyper-parameters obtained by tuning on the ResNet50 benchmark. We thus utilize common values for a dataset, effectively rendering them constants. The time taken to obtain these constants is thus a one-time cost, and does not impact the speedups obtained by LoCal+SGD.

Table 4: Analyzing impact of a common set of hyper-parameters

| Network | Training Strategy | Top-1 err. | Speed-Up |
|---|---|---|---|
| ResNet34 | LoCal+SGD (fine tuned) | 26.93% | 1.32× |
| | LoCal+SGD (common constants) | 27.04% | 1.26× |
| MobileNetV2 | LoCal+SGD (fine-tuned) | 28.82% | 1.36× |
| | LoCal+SGD (common constants) | 29.94% | 1.31× |

## 6.3 IMPACT OF WEAK SUPERVISION

In Table 5, we highlight the impact of the weak supervision technique on final classification accuracy. As can be seen, across all our benchmarks, the weak supervision technique clearly improves accuracy by nearly 0.06%-0.17%, bringing the final accuracy of LoCal + SGD closer to baseline SGD.

Table 5: Impact of Weak Supervision on accuracy

| Dataset | Network | Top-1 err. with weak supervision | Top-1 err. without weak supervision |
|---|---|---|---|
| ImageNet | ResNet34 | 27.04% | 27.1% |
| | ResNet50 | 24.41% | 24.49% |
| | MobileNetV2 | 28.94% | 29.03% |
| Cifar10 | VGG13 | 7.25% | 7.39% |
| | ResNet18 | 6.23% | 6.41% |
| Cifar100 | VGG13 | 31.59% | 31.7% |
| | ResNet18 | 23.63% | 23.75% |

## 6.4 ADDITIONAL COMPARATIVE ANALYSIS

In addition to the experiments performed in Section 3 to compare the performance of LoCal+SGD against existing techniques such as pruning during training (Lym et al., 2019) and training with stochastic depth (Huang et al., 2016), we conduct additional experiments to further solidify the superiority of our approach. We elucidate upon these comparisons as follows.

### 6.4.1 COMPARING LoCal + SGD AGAINST SGD AT ISO-ACCURACY

We compare the proposed LoCal+SGD training strategy against a SGD baseline that is trained with fewer epochs, i.e., the number of epochs required to reach the highest accuracy obtained by LoCal + SGD across the total training periods listed in Section 6.1. For the ImageNet benchmarks, the runtime improvements are listed in Table 6 below. Clearly, LoCal+SGD continues to achieve significant speed-ups (around 1.25×) compared to the SGD baseline, even for complex benchmarks such as ResNet50 and MobileNetV2.

### 6.4.2 COMPARING LoCal+SGD AGAINST FREEZING LAYERS DURING TRAINING

In Section 3, we compare LoCal+SGD against a technique, freezing layers during training, wherein instead of updating the layers using localized learning, the weights are held fixed. In this section,

Table 6: Analyzing LoCal+SGD speed-up at iso-accuracy

| Network | Training Strategy | Top-1 err. | Speed-Up |
|---------|-------------------|------------|----------|
| ResNet50 | SGD baseline (76 epochs) | 24.40% | 1× |
| | LoCal+SGD (83 epochs) | 24.41% | 1.26× |
| MobileNetV2 | SGD baseline (138 epochs) | 28.92% | 1× |
| | LoCal+SGD (146 epochs) | 28.94% | 1.24× |

we perform a more thorough comparison of LoCal+SGD against freezing layers during training. Specifically, we perform this comparison at iso-runtime, and analyze the resulting accuracy of either approach. To elaborate, we first identify the LoCal+SGD configuration that can reach the best accuracy within 0.05%, 0.1%, 0.25%, 0.5% and 1% of the baseline SGD accuracy. Then, for the same runtimes taken by each LoCal+SGD configuration, we identify the configuration that provides the best accuracy for the freezing layers approach. Our results for the Cifar10 ResNet18 benchmark can be found in Table 7. LoCal+SGD performs superior to freezing layers during training on 3 out of the 5 configurations studied, i.e., is a superior technique when the loss compared to SGD is allowed to exceed 0.1%.

Table 7: Comparing LoCal+SGD against freezing layers during training

| Maximum loss in Top-1 Accuracy | Loss in accuracy for LoCal+SGD | Loss in accuracy for freezing layers during training |
|--------------------------------|--------------------------------|------------------------------------------------------|
| 0.05% | 0.08% | 0.04% |
| 0.1% | 0.1% | 0.09% |
| 0.25% | **0.21%** | 0.28% |
| 0.5% | **0.41%** | 0.56% |
| 1% | **0.85%** | 0.97% |

### 6.5 ANALYSIS OF STATIC SCHEDULES FOR LEARNING MODE SELECTION

The current LoCal+SGD framework is realized with the help of an automatic learning mode selection algorithm, which determines the position of the transition layer every epoch. Instead of a dynamic data-dependent algorithm, we investigate the benefits of using a static schedule - that is, the position of the transition layer is determined using some pre-defined scheduling function. To this end, we have implemented a simple static schedule that favors aggressive application of the localized learning rule in initial layers, and gradually decreases the number of epochs localized learning is applied in the deeper layers. As shown in Equation 3, we opt for a quadratic scheduling function, as we empirically observe they perform better compared to the linear functions studied. Here N determines the position of the transition layer every epoch, $E_{max}$ is the maximum number of training epochs, and $c_1$ and $c_2$ are constants obtained using grid search.

$$N = \lfloor max(0, c_1 - c_2 \cdot (E - E_{max})^2) \rfloor \tag{3}$$

We report the results using this static schedule in Table 8 for the ImageNet-ResNet50 and MobileNetV2 benchmarks. Compared to the results reported in Table 1, we find that the static schedule achieves slightly higher runtime benefits, for marginally lower accuracies. However, static schedules suffer from some drawbacks – several static scheduling functions are feasible, e.g. exponential, quadratic, etc., and identifying the best scheduling function for each network requires extensive empirical analysis. The learning mode selection algorithm utilized in the paper helps alleviate this by automatically identifying the position of the transition layer every epoch, leveraging the gradient information at the boundary between localized updates and SGD.

Table 8: Analyzing Impact of Static Learning Mode Selection Schedule

| Network | Training Strategy | Top-1 err. | Speed-Up |
|---------|-------------------|------------|----------|
| ResNet50 | LoCal+SGD (static schedule) | 24.51% | 1.41× |
| MobileNetV2 | LoCal+SGD (static schedule) | 28.90% | 1.33× |

