# OpenReview forum: "Accelerating DNN Training through Selective Localized Learning "
_ICLR.cc/2021/Conference — Reject_

### Official Review · AnonReviewer2 · 2020-10-23
**First competitive results for a (partial) localized learning procedure. Has wide applications across many fields.**

**Rating:** 7
**Confidence:** 4

**Review:**

EDIT: 2020-11-27: Updated my score. Further explanations at the end of this post.

Summary:
The authors propose an algorithm that separates a network into two distinct regions where one is trained with SGD while the other is trained with a local Hebbian learning rule. A weak supervision rule is proposed that improves localized learning. The authors demonstrate close to baseline performance on CIFAR-100 and ImageNet for diverse networks while speeding up training.

Strong points:
- Very creative use of multiple learning rules during training.
- People could never get localized learning to work well enough to compete with SGD. This is the first work that uses some localized learning and manages to come close. This is a big success.
- These findings have broad applicability, from brain-like learning algorithms, efficient training on one GPU, and efficient learning algorithms for model parallelism for massive networks where communication is the bottleneck.

Weak points:
- Missing ablation for the weak supervision algorithm.
- This work is very impactful in many different ways, but it only mentions the computational efficiency perspective. The related work needs to be expanded to make readers aware of the impact of this work.
- The heuristic for learning mode selection is complicated. Initial experiments suggested that a simple learning-rate-schedule-like way to select the learning mode would be possible.
- Some algorithmic details in the weak supervision algorithm not clear.

Recommendation (short):
This is very important work with results that will significantly impact many fields (efficient training, parallelization, brain-like algorithms). It is a creative solution to a significant problem in localized learning. I strongly recommend accepting this work. If this work is rejected, I will no longer review for future ICLR conferences. I recommend this work to be accepted as an oral presentation. Selecting this work for a best paper award would be appropriate.

Recommendation (long):
This work is impactful for multiple reasons:
- Localized learning is difficult. There has not been a work that uses localized learning and makes it work close to SGD performance on large datasets/models.
- The brain does not use SGD, and it is difficult to think about algorithms that work in the brain and yield good performance. It might be that initial learning in the brain is done differently until local learning rules are used. This view is mostly ignored, but this paper yields evidence that such learning might be possible and efficient.
- Layers updated with localized rules can be updated independently of other layers (if the weakly supervised rule is not used). This enables fully asynchronous training for early layers. There will be a synchronization point at the SGD layers, but through pipeline parallelism, the communication overhead can be hidden in Hebbian layers. This enables the training of massive neural networks with trillions of parameters. With current parallelism tools becoming more and more limited, this is a crucial innovation since previous asynchronous parallelism procedures always decrease predictive performance. This is the first work that shows a way to do asynchronous parallel training without performance degradation.

Beyond this, the paper also yields some speedups for tasks while decreasing predictive performance only slightly. This is also an impressive feat, but the overall broad insights this paper yields make it much more impactful than this result. As such, I do not view the experimental results as the main contribution, but overall, the papers' main contribution is that it shows a way to include (gradual) localized learning in a neural network while not impacting performance.


Comments for authors:
This is excellent work — well done! I think the main weakness is currently a missing ablation on the effect of the weak supervision rule. You note that improves performance but by how much would be an important detail. If you do not include these ablations, I would still accept the paper, but I might rescind my oral presentation recommendation.

Another issue is that your work is relevant in many different domains, but you keep it confined to the idea that your method is only useful for faster training. I think making the reader aware that local training has many advantages across many domains could be very valuable. You do not need to elaborate on this, but I would like to see some of these connections in the paper because not everyone has the background to see these connections. I think you can do this mostly by mentioning it in the conclusion since you already mention a little bit of work in parallelism, and you mention previous results about local learning that failed to obtain good performance. Another line of work that I would mention in the related work section is that of neuroscientific faithful learning rules. The most relevant line of research is the work on various forms of feedback alignment and other algorithms. For a summary of past research and results on large datasets, see Bartunov et al., 2018[1]. Beyond this, you might want to add "sparse training" to the related work on efficient deep learning. Sparse training differentiates from pruning during training by initializing the neural network sparsely during initialization (not densely and then prune to sparse). See work on a mixture of experts[2,3] and sparse dynamic training[4,5,6]. I do not require you to include these references, but they might improve the related work section.

On the algorithmic and experimental side, it seems that a simple learning-rule-like schedule might be sufficient for selecting the learning mode. While I do not require you to add these experiments, it would make the algorithm simpler and more appealing if you can show that a simple learning rule works a la "warmup with SGD for 5 epochs, then shift by 1 layer (block) every 5 epochs" etc.

[1] Sartunov et al., 2018. Assessing the Scalability of Biologically-Motivated Deep Learning Algorithms and Architectures
[2] Shazeer et al., 2017. Outrageously Large Neural Networks: The Sparsely-Gated Mixture-of-Experts Layer
[3] Lepikhin et al., 2020. GShard: Scaling Giant Models with Conditional Computation and Automatic Sharding
[4] Mostafa & Wang et al., 2019. Parameter Efficient Training of Deep Convolutional Neural Networks by Dynamic Sparse Reparameterization.
[5] Dettmers & Zettlemoyer et al., 2019. Sparse Networks from Scratch: Faster Training without Losing Performance.
[6] Evci et al., 2019. Rigging the Lottery: Making All Tickets Winners.


Update:
==================================

Comments for Area Chair and Reviewers:

If I view this work merely by the story conveyed in the paper, my assessment would be more in line with the other reviewers. I am not quite sure if this is the right way to evaluate this paper since I view it as having a broader impact that goes beyond the story in the paper, but other reviewers disagree with my view on its broader impacts. I see this as a sign that the paper is currently not in a good enough state to really convey its potential impact.

Comments for authors:

I believe you still did good work here on the merits of the "speedup training" story that you convey in your paper. I believe that you have much more than this in your hands though. I think you could go two ways from here: (1) get this paper accepted in this form and work closely on the other angles that this work offers in a new paper, e.g. learning which is in line with biological or efficient parallelization of large networks. The second way (2) would be to rewrite this paper more in line with that view and resubmit. I think (1) might be better for you. I do not think many reviewers would understand a paper that comes from the process in (2). Good luck!

---

> ### Author Response · Authors · 2020-11-20
> **Response to Reviewer 2**
>
> We sincerely thank the reviewer for the positive feedback! We have incorporated the reviewer’s suggestions and have provided our response as follows.
>
>
> 1.	Weak Supervision Ablation Analysis:
>
> We have included an analysis of the impact of weak supervision for the ImageNet benchmarks in Table 5 (page 13) of the revised paper. As can be seen, the impact of the weak supervision technique on accuracy is 0.06-0.17%. Removing weak supervision leads to only a minimal degradation in accuracy, allowing us to successfully realize model parallelism during training as pointed out by the reviewer.
>
> 2.	Expanding Related Work:
>
> We have updated the related work sections to include bio-plausible learning rules such as Target Propagation and Feedback Alignment. Due to space limitations, we are unfortunately unable to include references to sparse training efforts. It is definitely an interesting suggestion and will certainly be included with the increased page limit if the paper is accepted.
>
> 3.    Static learning rate like schedule for learning mode selection :
>
> As suggested by the reviewer, we have implemented a simple static schedule to determine the position of the Localized->SGD transition layer every epoch. We have updated the paper to include such an analysis in Section 6.5. The static schedule favors aggressive application of the localized learning rule in initial layers, and gradually decreases the number of epochs localized learning is applied in the deeper layers.
>
> N= [max⁡(0,$c_{1}$- $c_{2}$*(E- $E_{max}$ )^2 ]
>
> We opt for a quadratic scheduling function, as we empirically observe they perform better compared to the linear functions studied. Here N determines the position of the transition layer every epoch, $E_{max}$ is the maximum number of training epochs, and $c_{1}$ and $c_{2}$  are constants obtained using grid search.
>
> We report the results using this static schedule in Table 8 (page 14) of the revised paper, on the ImageNet-ResNet50 and MobileNetV2 benchmarks. Compared to the existing approach used in the paper (Table 1,page 6), we find that the static schedule achieves slightly higher runtime benefits, for marginally lower accuracies. However, static schedules suffer from some drawbacks – several static scheduling functions are feasible, e.g. exponential, quadratic, etc., and identifying the best scheduling function for each network requires extensive empirical analysis. The learning mode selection algorithm utilized in the paper helps alleviate this by automatically identifying the position of the transition layer every epoch, leveraging the gradient information at the boundary between localized updates and SGD.

---

### Official Review · AnonReviewer1 · 2020-10-28
**A possible useful training practice, but some important experiments are missing**

**Rating:** 6
**Confidence:** 4

**Review:**

This paper try to leverage the benefit of Hebb learning to reduce CNN training time cost. In order to achieve this,  a learning mode selection algorithm is proposed to progressively increase  number of layers using Hebb learning. The writing of this paper is good and the idea is also interesting, however, the experimental part should be improved:

1. The criterion used in learning mode selection algorithm is the model-update norm of current epoch. If the norm is small enough, the transition layer index will be increased. A small model-update norm also means that current layer is nearly convergent. Could you just fix these layers to accelerate training? Yes, freezing layer exps are tried but the comparison is not fair in my opinion. When Hebb-Learning-Layers are frozen, the final accuracy drops, but the training speedup is improved. So if you freeze less layers to make training speedups of freezing-layer-training and Hebb-Learning same, what will the accuracy relationship be? Does the proposed method still outperforms freezing strategy?

2. A weak supervision scheme is proposed in this paper, but I did not find any experiments to evaluate its effect, could you add this part?

---

> ### Author Response · Authors · 2020-11-20
> **Response to Reviewer 1**
>
> We thank the reviewer for the suggestions, please find our response below.
>
> 1. Comparison against freezing layers during training :
>
> As requested by the reviewer, we perform a more thorough comparison of LoCal+SGD against freezing layers during training. Specifically, we perform this comparison at iso-runtime, and compare the resulting accuracies of the approaches. We have updated the paper to include this analysis in Section 6.4.2.
>
> We first identify the best LoCal+SGD configuration that can reach within 0.05%, 0.1%, 0.25%, 0.5% and 1% of the baseline SGD accuracy. Then, for the runtimes taken by each LoCal+SGD configuration, we identify the number of layers that must be frozen to match the training time, and report the resulting accuracy. Our results for the Cifar10 Resnet18 benchmark can be found in Table 7, page 14 of the revised paper. LoCal+SGD performs superior to freezing layers during training on 3 out of the 5 configurations studied, i.e., is a superior technique when the loss compared to SGD is allowed to exceed 0.1%.
>
>
> 2. Weak Supervision Evaluation :
>
> We have included the requested analysis in Table 5, page 13 of the revised paper. As seen, the weak supervision technique adds around 0.06-0.17% increase in Top-1 accuracy.

---

### Official Review · AnonReviewer4 · 2020-10-28
**Needs more work**

**Rating:** 5
**Confidence:** 4

**Review:**

This work demonstrates that localized learning can improve DNN training efficiency by reducing computation and memory requirements. The effectiveness of this approach is shown by the experimental results that report a nice trade-off of minimal accuracy loss and good throughput improvement compared to baseline SGD and competing efficient training techniques.
While the experimental results are indeed appealing, a major flaw is that the paper does not sufficiently explain why the underlying techniques (learning mode selection and weak supervision) work. Since these techniques are configured with a number of hyper-parameters, and I could not gain an intuition of how/why/whether they work in general. For example, $t\_{shift}$ is set to recurring block size of residual nets, but what is not given is the justification for this choice, or how to set for non-residual nets. In other words, the hyperparameter settings for these techniques appear ad-hoc (I suspect that it is not), and so it is not clear to me how much exploration is required. What could have helped is to include a study of the incremental benefits of these techniques in the experiment section. In summary, while the results are good, the writing and presentation could be greatly improved to help readers learn and use the proposal.
`
**Pros**
1. Tackles an important problem of reducing time and resource requirements of DNN training.
2. The general approach of computing layers differently over the course of training is quite intuitive.
3. Presents two techniques that appear to make localized learning practical and effective for DNN training.

**Cons**
1. The proposed techniques are parameterized (e.g., $t\_{shift}$, $\alpha$, etc.), but how, and effort required to configure them is not clear.
2.  The proposed techniques are not sufficiently explained to help build intuition. For example, the weak supervision suggests that reversing the weight update direction can effectively reverse increase in classification loss (i.e., divergence), but it is not clear why this is the case or where this observation applies to SGD and other optimizers.
3. Incremental benefits of the techniques is not provided in evaluation.

---

> ### Author Response · Authors · 2020-11-20
> **Response to Reviewer 4**
>
> We thank the reviewer for the suggestions, please find our response below.
>
> 1. Effort to configure hyper-parameters
>
> As pointed out to Reviewer 5, we first tune the hyper-parameters $\alpha$, $t_{shift}$, $L_{max}$   using simple grid search on a single benchmark for a particular dataset, and utilize the same hyper-parameters across all networks trained on that dataset. This vastly reduces time spent in hyper-parameter tuning, rendering it as merely a one-time cost. Moreover, to justify our use of a common set of hyper-parameters, we have listed the accuracy and runtime benefits accrued by tuning the $\alpha$, $t_{shift}$, $L_{max}$ parameters for each individual network in Table 4, page 14 of the revised paper– what we observe is only a marginal improvement in accuracy and runtime benefits, over using a common set of hyper-parameters. We have updated the paper to include this analysis in Section 6.2.
>
> In Figure 6 of Section 3.3 of the revised paper, we have conducted ablation studies for the  $t_{shift}$ and $L_{max}$ parameters for the ResNet50 network trained on the ImageNet dataset. The ablation analysis performed for $t_{shift}$  indicates that the accuracy benefits are highest for $t_{shift}$  in the range of 5-10% of total network depth, and is largely constant within this range. Further, the runtime benefits are largely insensitive to $t_{shift}$  as the average number of layers updated locally remain the same. We hence set $t_{shift}$ to a size of a recurring block for MobileNets and ResNets (For ResNet50 and MobileNetV2, $t_{shift}$≈6% of network depth) . For VGG-like networks, it is set to a single convolutional layer.
>
> For $L_{max}$, it is clear from the same figure that accuracy begins to drop steeply when localized updates are applied beyond 75-80% of the network layers. We hence set to $L_{max}$ to 75% in all our experiments. As mentioned previously, we find that these values of $t_{shift}$ and $L_{max}$ translate well to other benchmarks such as MobileNetV2 and ResNet34.
>
>
> 2.   Weak Supervision Technique:
>
> The weak supervision technique introduced is a coarse-grained, hyper-parameter free, low-cost supervision mechanism that modulates the learning rate of the locally updated layers, depending on the changes in the global classification loss. As mentioned in the paper, whenever the global classification loss increases, weak supervision encourages the weight updates of the locally updated layers to proceed in the reverse direction. The additional accuracy achieved by the weak supervision technique, on top of the learning mode selection algorithm, is around 0.06-0.17% across all our benchmarks- this boost in accuracy brings the LoCal+SGD training technique closer to the baseline SGD in terms of accuracy.
>
> Further, we note that we studied different weak supervision techniques, which for example rewarded/penalized the weight updates by different magnitudes when the classification loss decreased/increased. Such techniques provided a higher boost in accuracy (~1%) compared to the existing technique in the paper. However, due to the added hyper-parameter complexity, we opted for the simple, hyper-parameter free technique presented in the paper.
>
> Finally, we point out that the weak supervision technique developed will not be beneficial for optimizers such as SGD/Adam, as the errors are already calculated in a fine-grained manner for each neuron, based on overall classification loss.
>
> 3. Incremental Benefits
>
> We assume that the reviewer is referring to the incremental benefits of the two techniques, i.e., learning mode selection algorithm and weak supervision. We have thus listed the accuracy with and without using weak supervision, in Table 5 (page 13) of the revised paper.

---

### Official Review · AnonReviewer3 · 2020-10-29

**Rating:** 4
**Confidence:** 4

**Review:**

This paper proposes a combination of SGD with selective application of a non-backprop learning rule (Hebbian). The two learning rules are not applied together, but rather a boundary is determined where layers prior use SGD, and the ones after use the Hebbian approach. A selection algorithm dynamically adjusts the boundary over training. For accuracy reasons, they include weak supervision by using the overall classification loss to control the sign of the update.

From computational efficiency perspective, the contributions reduce the need for a backprop calculation, and also leads to a smaller memory footprint, since the activation values need not be stored. On ImageNet benchmark models, they show <0.5% Top1 drop in exchange for ~1.3x runtime speed compared to vanilla SGD.

Strengths
+ Focus on run-time improvements brings practical significant to their proposed method
+ Algorithm is relatively simple to implement
+ Convergence results only show a small degradation from SOTA
+ The tuning results for alpha (Figure 5) are useful for practitioners that need to balance accuracy and compute

Weaknesses
- The 'meta' boundary selection and weak supervision approaches add additional hyperparameter complexity to the tuning process. These are also empirically but not theoretically motivated, and unclear if they generalize to other domains. I understand that non-classification models are out of scope for this paper, but his paper's impact would be improved by some comment on transferability. For example, U-net models have long range skip connections that span the model.
- While the focus on runtime is welcome, what is relevant to the practitioner is time-to-train to a particular accuracy target, similar to the metric adopted by MLPerf. Since this method does introduce an accuracy degradation (e.g. for RN-34, 27.04% versus 26.6%; Table 1), the more fair comparison would account for the fewer epochs the baseline SGD needs to hit 26.6%. To make a more convincing argument to practitioners, I would compare either the wall-clock time, or the total FLOPS needed, to hit the target accuracy.
- Several techniques are introduced without ablations to measure their effectiveness and justify the added complexity. This is particularly important as these techniques add additional burden on the practitioner in terms of tuning the new hyperparameters.

The work combines existing learning rules (SGD, Hebbian) with some novelty in how they are employed, and with a weak supervisory signal, to achieve reasonable results. These contributions were not foundational improvements, so the paper's main merit is in the potential practical impacts of this method. The significance to practitioners however, is greatly reduced by the weaknesses described above.

---

> ### Author Response · Authors · 2020-11-20
> **Response to Reviewer 3**
>
> 1.  Applicability to other domains
>
> As the reviewer pointed out, the primary focus of this work is on feed-forward classification models. While applying localized learning rules to smaller LSTM/RNN networks may be feasible, some additional thought is required in developing the hybrid training strategy, i.e., learning mode selection algorithm, in a manner that achieves a good trade-off between accuracy and training runtime. It is an interesting suggestion, and we will investigate it further as part of future efforts.
>
> However, extending LoCal+SGD to U-Nets is not difficult. The long-range connections are handled similar to the shortcut connections in ResNets. Consider a Layer K, whose input and output activations are $A_{K-1}$ and $A_{K}$. Further, let us assume Layer K receives activation input $A_{J}$ from a preceding layer J. The weight update for Layer K is performed by convolving the summed activation $A_{K}$ + $A_{J}$, with $A_{K-1}$. Based on the reviewer’s suggestion, we have conducted experiments that demonstrate the feasibility of LoCal+SGD to U-Net training. The results are demonstrated in Table A below. We utilize the U-Net architecture from [1], and the isbi challenge dataset from [2].
>
> $\hspace{3in}$  Table A
> ++++++++++++++++++++++++++++++++++++++++++++++++++++++++++++++++++++++++++++++++++++
> Benchmark           $\hspace{1.3in}$                                          Dice Coefficient [3] 	$\hspace{0.7in}$		                                 Speed-Up
> ++++++++++++++++++++++++++++++++++++++++++++++++++++++++++++++++++++++++++++++++++++
> SGD baseline 	$\hspace{1.3in}$       	                                     0.948	$\hspace{1.3in}$				                                 1x
> LoCal+SGD             $\hspace{1.4in}$                                             0.944      $\hspace{1.3in}$					                                1.28x
> ++++++++++++++++++++++++++++++++++++++++++++++++++++++++++++++++++++++++++++++++++++
>
> [1] U-Net: Convolutional Networks for Biomedical Image Segmentation: Olaf Ronneberger, Philipp Fischer and Thomas Brox
>
> [2] http://brainiac2.mit.edu/isbi_challenge
>
> [3] https://en.wikipedia.org/wiki/S%C3%B8rensen%E2%80%93Dice_coefficient
>
> 2. Comparison against SGD baseline with fewer epochs:
>
> As requested by the reviewer, we compare LoCal+SGD against an SGD baseline that is trained with fewer epochs, i.e., at iso-accuracy. We have updated the paper with this analysis in Section 6.4.1 for the ImageNet benchmarks. Clearly, LoCal+SGD achieves runtime benefits even when compared against a baseline with fewer training epochs.
>
>
> 3.  Ablation analysis
>
> We thank the reviewer for the suggestion. We have conducted an ablation analysis for $t_{shift}$ and $L_{max}$ and updated the paper in Section 3.3.
>
> The ablation analysis performed for  $t_{shift}$  indicates that the accuracy is highest for $t_{shift}$  in the range of 5-10% of total network depth, and is largely constant within this range. The impact of $t_{shift}$ on runtime benefits is quite straightforward – the runtime benefits are largely insensitive to $t_{shift}$, as the average number of layers updated locally remains nearly the same across the training period. In all our experiments, we thus set $t_{shift}$ to 5-10% of total network depth.
>
> For $L_{max}$, it is clear from the same figure that accuracy begins to drop steeply when localized updates are applied beyond 80% of the network layers. Additionally, the runtime benefits naturally scale with increasing values of $L_{max}$.  Thus, $L_{max}$ is set to 75% in all our experiments.
>
> Furthermore, as pointed out to Reviewer 5, the above hyper-parameters are first identified using grid-search on one network, and the same values are used across all networks of a particular dataset. The impact of using such common hyper-parameter values, instead of tuning on each individual network is minimal, as shown in Table 4 in the revised version of the paper. We thus reduce hyper-parameter tuning to a one-time cost.

---

### Official Review · AnonReviewer5 · 2020-11-04
**interesting work but needs more details**

**Rating:** 6
**Confidence:** 4

**Review:**

Accelerating DNN Training through Selective Localized Learning

In this paper, the authors proposed a new approach by the name of LoCal + SGD (Localized Updates) to replace the traditional Backpropagation method. The key idea is to selectively update some layers’ weights using localized learning rules. For these layers, the computation cost is reduced from two matrix multiply operations to one matrix multiply operation. The authors also proposed the Learning Mode Selection Algorithm to maintain the accuracy and convergence.

The authors provided some experimental results on common deep learning benchmarks such as ImageNet/ResNet and CIFAR/VGG. Overall, the authors reported that this approach can achieve around 1.36x speedup for a 0.4% loss in accuracy for ResNet-50. The authors also reported that they can achieve a higher speed than recent methods such as Structured Pruning and stochastic depth.

This work is a trade-off between computation and accuracy (details in Figure 5). I have some questions for the authors:

(1) How stable is the proposed method (LoCal + SGD)? Does it still work for large-batch optimization and asynchronous training?

(2) What is the overhead of hyper-parameter tuning?

(3) Did the authors use the same number of epochs as the baseline to finish the training?

(4) What is the absolute speed (e.g. in GFlops or TFlops)? Can the proposed method beat a well-optimized NVIDIA implementation?

(5) What is the limit of the proposed method?

(6) Can the Learning Mode Selection Algorithm work with other methods?

Since this work fundamentally changes the way of learning, it is probably necessary to do a convergence analysis for the proposed method.

---

> ### Author Response · Authors · 2020-11-20
> **Response to Reviewer 5, Part 1/2**
>
> We thank the reviewer for the helpful comments, please find our response below.
>
> 1.	Application of LoCal+SGD to large batch training:
>
> To demonstrate the applicability of LoCal+SGD to large batch training, we ‘simulate’ a batch size of 32k on 4 GPUs, by using gradient accumulation [1]. The results are as follows in Table A, for the ResNet50 network trained using ImageNet:
>
>  $\hspace{3in}$  Table A
> ++++++++++++++++++++++++++++++++++++++++++++++++++++++++++++++++++++++++++++++++++++
> Benchmark           $\hspace{1.3in}$                                          Top-1 Error % 	$\hspace{0.7in}$		                                 Speed-Up
> ++++++++++++++++++++++++++++++++++++++++++++++++++++++++++++++++++++++++++++++++++++
> SGD baseline 	$\hspace{1.3in}$       	                                     26.8	$\hspace{1.3in}$				                                 1x
> LoCal+SGD             $\hspace{1.4in}$                                             27.24`      $\hspace{1.3in}$					                                1.33x
> ++++++++++++++++++++++++++++++++++++++++++++++++++++++++++++++++++++++++++++++++++++
>
>
> As can be seen, LoCal+SGD remains scalable in a massively distributed scenario.
> [1]: https://pytorch-lightning.readthedocs.io/en/latest/training_tricks.html
>
>
> 2.   Overhead of hyper-parameter tuning:
>
> To realize LoCal+SGD, we introduce three hyper-parameters: $\alpha$, $t_{shift}$ and $L_{max}$. $t_{shift}$ controls the number of layers that switch to SGD-based updates every epoch, $L_{max}$ is the maximum number of layers that can be use localized learning, and $\alpha$ defines a threshold that is used to determine whether the transition layer is shifted deeper into the network.
>
> To obtain optimized values for these hyper-parameters, we first perform a grid search using a single network for a particular dataset (for example, we choose the ResNet50 network for ImageNet). We transfer the same hyper-parameter values to all other networks for the same dataset. We justify our use of common hyper-parameter values by the following experiment. In Table 4 (page 13) we depict the results on other ImageNet benchmarks (ResNet34 and MobileNetV2) when hyper-parameter tuning is performed for each benchmark individually. As can be seen, the accuracy and runtime benefits are only marginally better than those obtained using a common set of hyper-parameters obtained by tuning on the ResNet50 benchmark. We thus utilize common values for a dataset, effectively rendering them constants. The time taken to obtain these constants is thus a one-time cost, and does not impact the speedups obtained by LoCal+SGD.
>
> We have included the above analysis in Section 6.2 of the paper.
>
> 3. Number of epochs used for training:
>
> The number of epochs used for LoCal+SGD is the same as that of the baseline. We have clarified this in Section 6.1 (Appendix) of the revised manuscript.
>
> 4. Absolute speed, implementation details :
>
> We compare the LoCal+SGD implementation against optimized CUBLAS libraries (CUDA version is 9.0). The absolute speed i.e., average time taken by LoCal+SGD per epoch, to process a minibatch of size 64 on a single Nvidia GTX 1080 Ti GPU for the ImageNet benchmarks are reported in Table B below.
>
>  $\hspace{3in}$  Table B
> ++++++++++++++++++++++++++++++++++++++++++++++++++++++++++++++++++++++++++++++++++++
> Benchmark           $\hspace{1.3in}$                                          Total Time Taken to Process 1 minibatch=64 images
> ++++++++++++++++++++++++++++++++++++++++++++++++++++++++++++++++++++++++++++++++++++
> ResNet34 	$\hspace{2in}$       	                                    0.0128 sec
> ResNet50           $\hspace{1.95in}$                                             0.0247 sec
> MobileNetV2      $\hspace{1.8in}$                                            0.0121 sec`
> ++++++++++++++++++++++++++++++++++++++++++++++++++++++++++++++++++++++++++++++++++++
>
> 5.  Limitations of proposed efforts :
>
> We have discussed the limitations of our work in Section 3.3 of the revised manuscript. LoCal+SGD is difficult to apply to the deeper layers of the network – aggressive application of localized learning rules to more than 75-80% of the network layers often leads to severe losses in accuracy. Further, the applicability of such an approach to LSTMs, Transformers, and memory-augmented neural networks is also an open challenge that would be interesting to explore as part of future efforts.

---

> > ### Author Response · Authors · 2020-11-20
> > **Response to Reviewer 5, Part (2/2)**
> >
> > 6. Applicability to other methods:
> >
> > We are not sure what the reviewer is referring to as “other methods”. We assume the reviewer is referring to other optimization algorithms, such as Adam. For the paper, we use SGD with Nesterov momentum as the optimization method. To address the reviewer’s question, we also implemented hybrid LoCal+Adam training for the Cifar10-ResNet benchmark, and present our results in Table C below. The results suggest that the approach can also work with other optimizers.
> >
> >  $\hspace{3in}$  Table C
> > ++++++++++++++++++++++++++++++++++++++++++++++++++++++++++++++++++++++++++++++++++++
> > Training Technique           $\hspace{1.3in}$                                       Top-1 Error % 	$\hspace{0.7in}$		                                 Speed-Up
> > ++++++++++++++++++++++++++++++++++++++++++++++++++++++++++++++++++++++++++++++++++++
> > Adam 	$\hspace{2.3in}$       	                                     7.7	$\hspace{1.3in}$				                                 1x
> > LoCal+Adam             $\hspace{1.9in}$                                            7.96`    $\hspace{1.2in}$					                                1.48x
> > ++++++++++++++++++++++++++++++++++++++++++++++++++++++++++++++++++++++++++++++++++++
> >
> >
> >
> > 7. Convergence Analysis:
> >
> > We thank the reviewer for the suggestion. While a convergence analysis is a valuable suggestion, we would like to note that LoCal+SGD falls into the bucket of empirical techniques that accelerate training such as [1] and [2]. Theoretical analysis is out of the scope of this paper, but we will certainly consider it as part of future efforts.
> >
> > [1] Deep Networks with Stochastic Depth: Gao Huang, Yu Sun, Zhuang Liu, Daniel Sedra and Kilian Q. Weinberger
> > [2] E2-Train: Training State-of-the-art CNNs with Over 80% Energy Savings: Yue Wang, Ziyu Jiang, Xiaohan Chen, Pengfei Xu, Yang Zhao, Yingyan Lin, Zhangyang Wan

---

> > > ### Comment · AnonReviewer5 · 2020-11-25
> > > **Thanks so much for your efforts**
> > >
> > > I carefully read the rebuttal and really appreciate the efforts of the authors.
> > >
> > > Pros:
> > >
> > > This is an interesting idea. It may inspire researchers to create fundamentally new methods to replace Backpropagation.
> > >
> > > Cons:
> > >
> > > The generalization performance is not good, especially for large-batch optimization and LoCal+Adam.
> > >
> > > The trade-off is accuracy and speed. It may hurt the convergence for some applications.
> > >
> > > The limitation of the proposed method may be a concern.
> > >
> > > Therefore, I'd like to keep my current rating.

---

### Decision · Program_Chairs · 2021-01-07
**Final Decision**

**Decision:**

Reject

**Comment:**

In this paper, the authors proposed a new approach by the name of LoCal + SGD (Localized Updates) to replace the traditional Backpropagation method. The key idea is to selectively update some layers’ weights using localized learning rules, so as to reduce the computational complexity of training these layers so as to achieve a better tradeoff between overall speed and accuracy.
The paper received quite mixed reviewers. Some reviewers criticized the incremental nature of the proposed technology, while some other reviewers thought that this is one of the very early papers that demonstrates the practical effectiveness of localized learning.

The reviewers have made several rounds of discussions, and as a result of that, we think while this direction (localized learning) is very important and promising, this particular paper might not have provided a sufficiently novel and good solution to it.  Specifically, in terms of localized learning, this paper has not proposed brand new concepts or methodologies, instead it adopts existing methods in selective layers. In this sense, it does not really resolve the accuracy issue of localized learning, rather, it achieves the tradeoff by only applying localized learning in some layers. In other words, the current results still heavily rely on BP and has not brought a real breakthrough to localized learning.